# Identification of Critical Candidate Genes Controlling Monokaryon Fruiting in *Flammulina filiformis* Using Genetic Population Construction and Bulked Segregant Analysis Sequencing

**DOI:** 10.3390/jof11070512

**Published:** 2025-07-08

**Authors:** Peng Wang, Ya Yu, Lei Xia, Qi Yan, Xiao Tan, Dongyin Wang, Xue Wang, Zhibin Zhang, Jiawei Wen, Xiao Huang

**Affiliations:** 1Jilin Academy of Agricultural Sciences (Northeast Agricultural Research Center of China), Changchun 130033, China; wangpeng@cjaas.com (P.W.); yuya@cjaas.com (Y.Y.); xialei@cjaas.com (L.X.); tanxiao@cjaas.com (X.T.); 2College of Horticulture, Jilin Agricultural University, Changchun 130118, China; 20231362@mails.jlau.edu.cn; 3Institute of Science and Technology Information of Jilin Province, Changchun 130033, China; jlkjqk@163.com (D.W.); wangxue.happy@163.com (X.W.); 4School of Life Science, Northeast Normal University, Changchun 130024, China; zhangzb554@nenu.edu.cn

**Keywords:** *Flammulina filiformis*, genetic population construction, BSA-seq, monokaryotic fruiting

## Abstract

Fruiting body formation in edible fungi is a critical development process for both scientific understanding and industrial cultivation, yet the underlying genetic mechanisms remain poorly elucidated. This study aimed to identify key genes regulating monokaryotic fruiting in *Flammulina filiformis*. A genetic segregation population was constructed through selfing purification and hybrid segregation of the FF002 strain, followed by mapping candidate genes with bulked segregant analysis sequencing (BSA-seq). A 10 kb genomic region on scaffold19 was identified, pinpointing the gene *FV-L110034160*, which encodes a U2 snRNP complex component involved in pre-mRNA splicing. A T→G SNP located 121 bp downstream of the ATG codon caused a serine-to-alanine substitution, disrupting a conserved domain and altering fruiting phenotypes. Phylogenetic analysis further revealed conservation of this gene in fungal genera. These findings elucidate a key regulatory gene controlling monokaryotic fruiting in *F. filiformis*, providing novel insights into fruiting body formation mechanisms and establishing a foundation for genetic studies in other edible fungi.

## 1. Introduction

The mechanism of fruiting body formation is both a fundamental scientific question for understanding the growth and development of edible fungi and a critical practical issue for their efficient domestication and cultivation. Extensive investigations have explored this from multiple perspectives, such as environmental factors, nutritional physiology, and mutant analyses, leading to the identification of numerous associated genes [1,2,3,4]. Notably, the rapid advancement and application of genome and transcriptome sequencing technologies in life sciences have significantly enhanced the efficiency of gene discovery. Notably, recent studies have employed multi-omics approaches to investigate the regulatory genes involved in fruiting body formation in various edible and medicinal fungi, including *Schizophyllum commune* [5], *Agaricus bisporus* [6], *Coprinus cinereus* [7], *F. filiformis* [8], *Ganoderma lucidum* [9], *Lentinula edodes* [10], *Hypsizygus marmoreus* [11], and *Cyclocybe aegerita* [12]. These studies revealed that upregulated genes are predominantly associated with signal transduction, RNA transcription, protein synthesis, carbohydrate metabolism, and cell wall biogenesis pathways, while downregulated genes were primarily involved in sugar transport and glycolysis pathways. However, the omics-based approaches often yield an overwhelming number of regulatory genes, making it significantly challenging to identify the key control genes that directly control fruiting body formation.

Bulked segregant analysis (BSA) is a method that utilizes phenotypic differences to construct segregating populations for rapid screening and mapping of genes associated with target traits [13]. This approach has been extensively applied in functional gene mapping studies in plants and animals [14,15], and is increasingly utilized in edible fungi research [16]. However, challenges such as difficulties in marker development and labor-intensive screening remain persist. BSA-seq, which integrates BSA with high-throughput sequencing, employs SNPs and InDels directly as markers. This method offers advantages including shorter experimental timelines, precise mapping, uniform marker distribution, and high marker density, establishing it as one of the most effective strategies for functional gene discovery [17,18,19]. For fungal gene mapping, Song employed BSA-seq to dissect the gene network regulating pseudohyphal growth in *Saccharomyces cerevisiae* and identified multiple loci associated with invasive growth, including known pseudohyphal growth genes *FLO8* and *FLO11*, as well as the negative regulator *SFL1* [20]. Using BSA-seq, Jared identified a putative xylitol dehydrogenase gene, *XDH1*, demonstrating that wild *S. cerevisiae* can use xylose as its sole carbon source for biomass accumulation [21]. Moreover, Swart performed BSA-seq to map mutation sites linked to metabolic or growth traits, demonstrating its applicability for asexual fungi [22]. Despite challenges such as dependency on reference genomes and complexities in population construction, BSA-seq remains a critical tool in fungal genetic research due to its precision and efficiency.

*F. filiformis*, commonly known as enoki or winter mushroom [23], is prized for its culinary qualities and delicate flavor. It exhibits a well-characterized life cycle, rapid growth rate, and readily obtainable sexual and asexual spores [24], enabling efficient purification of genetic backgrounds through genetic manipulation. Additionally, *F. filiformis* possesses monokaryotic fruiting capability and can form fruiting bodies in culture media [25,26,27]. This characteristic not only effectively mitigates the influence of environmental and nutritional differences on fruiting body development but also minimizes interference from mating-type genes and allelic variations, making *F. filiformis* an ideal model for studying the mechanisms underlying fruiting body formation in edible fungi.

Here, through genetic population constriction and BSA-seq, we identified key regulatory genes involved in fruiting body formation in *F. filiformis* and delineated their associated metabolic pathways. These findings not only advance our understanding of the molecular mechanisms governing fruiting body formation in *F. filiformis* but also provide valuable insights for studying fruiting body development in other edible fungi. Furthermore, this research has significant implications for developing novel domesticated varieties and promoting sustainable practices in the edible mushroom industry.

## 2. Materials and Methods

### 2.1. Strain Identification

The wild *F. filiformis* strain FF002 was isolated from poplar tree roots in Rongting Street, Nanguan District, Changchun City, Jilin Province, China (125°23′26″E, 43°49′29″N). Monokaryotic strains derived from FF002 are categorized into two distinct groups based on their fruiting capabilities: fruiting and non-fruiting (Figure 1).

### 2.2. Identification of Monokaryotic Fruiting Traits

The ventral surface of the FF002 mature fruiting body was placed face down in a sterile 9 cm Petri dish. After 3–6 h of static incubation at 25 °C, white spore prints were obtained by removing the fruiting bodies. The spore suspension mother liquor was made in a clean bench. The spore concentration was quantified using a hemocytometer and diluted to 100–200 spores/mL. A 100 μL aliquot of the suspension was evenly spread onto PDA medium using a sterile bent glass rod, and the plates were incubated at 25 °C. Spores germinated within 3–5 days. A single colony was picked out and inoculated into a test tube containing PDA medium and labeled. Monokaryotic strains were identified using double-fluorescence staining [28].

These strains were cultured at a constant temperature of 25 °C in the dark, and their monokaryotic fruiting status was recorded after 30 days. As the objective of this study was to identify genes controlling the transition from vegetative to reproductive growth in fruiting body formation in *F. filiformis*, rather than genes regulating fruiting body growth and development, monokaryotic strains that formed only primordia without further differentiation were also classified as exhibiting fruiting traits.

### 2.3. Genetic Population Construction

#### 2.3.1. Preparation of Homozygous Strains with Monokaryotic Fruiting and Non-Fruiting Traits

Monokaryotic strains of FF002, exhibiting either fruiting or non-fruiting traits, were self-crossed within each trait group to produce dikaryotic strains. After fruiting body formation, monokaryotic strains were reisolated from these dikaryons, and their fruiting capacities were assessed. Monokaryotic strains maintaining consistent trait types were selected for repeated cycles of (i) self-crossing, (ii) fruiting, (iii) monokaryon preparation, and (iv) fruiting trait assessment, until the monokaryotic progeny of self-crossed strains exhibited no trait segregation. This iterative process ultimately yielded near-isogenic self-crossed lines that differed exclusively in monokaryotic fruiting trait, resulting in homozygous dikaryotic strains with stable monokaryotic fruiting or non-fruiting traits (Figure 2).

#### 2.3.2. Construction of a Genetic Segregating Population for Monokaryotic Fruiting Traits

The two homozygous parental strains were crossed, and basidiospores were collected from the fruiting bodies of the hybrid strain. Monokaryotic strains derived from these spores were phenotypically evaluated for fruiting capability. This monokaryotic strain population represents a genetic segregation population for monokaryotic fruiting traits and serves as the mapping population for associated regulatory genes (Figure 2).

### 2.4. Genomic DNA Extraction

Inoculum blocks (5 mm × 5 mm) from strains of the genetic segregation population were inoculated into PD medium and incubated at 25 °C with shaking at 120 rpm under dark conditions for 10 days. The resulting mycelia were harvested, pressed with sterile filter paper to remove excess moisture, flash-frozen in liquid nitrogen, and stored at −80 °C for subsequent use. Genomic DNA was extracted using a QIAamp DNA Mini Kit (Qiagen, Hilden, Germany). The quality of purified genomic DNA was examined using a NanoDrop 2000 UV spectrophotometer (Thermo Scientific, Waltham, MA, USA) and a Qubit 2.0 fluorometer (Life Technologies, Carlsbad, CA, USA).

### 2.5. BSA-Sequencing

A total of 24 DNAs were selected from the two types of monokaryotic strains (fruiting and non-fruiting) and pooled separately. Whole-genome resequencing was performed on DNA extracted from the two bulked pools, including the monokaryotic fruiting FF002-F and the monokaryotic non-fruiting FF002-N, as well as two parental strains (FF002-PL-30F and FF002-PL-8N). Genomic DNA was sonicated, and the adapters were ligated to the fragment ends. After purification, the library was amplified for 150 bp paired-end sequencing on the Illumina X-Ten platform (Genedenovo Biotechnology Co., Ltd., Guangzhou, China). After removing the adapters and low-quality reads, the clean reads from each sample were aligned against the FV-L11 genome [29] using the Burrows–Wheeler Aligner (0.7.17) [30]. Then, alignment files were converted to BAM format using SAMtools (1.22) [31]. Variant calling was identified using the Genome Analysis Toolkit3 (3.8-1) [32], followed by single-nucleotide polymorphisms (SNPs) or insertion/deletion polymorphisms (InDels) annotation using ANNOVAR (http://annovar.openbioinformatics.org, accessed on 12 January 2025) [33]. Association analysis was performed using the SNP-index [34], Δ(SNP-index) [17], calculation of G statistic [35], Euclidean distance (ED) [36], and two-tailed Fisher exact test [37] based on the SNPs. Finally, the overlapping interval of the four methods was considered the final QTL interval.

### 2.6. Functional Annotation of Candidate Genes

The nucleotide and protein sequences of candidate genes within the mapped interval were retrieved according to genomic data and annotation information. Protein homology analysis and conserved domain annotations were performed using NCBI Blastp (https://blast.ncbi.nlm.nih.gov, accessed on 18 January 2025) and the Conserved Domains Database (CDD) (https://www.ncbi.nlm.nih.gov/Structure/cdd/cdd.shtml, accessed on 18 January 2025), respectively. SNP mutation within the coding sequences of the candidate genes was identified based on the BSA-seq results. Genes harboring missense mutations that altered the encoded protein within conserved domains, and showing significant association between mutation status (in both parents and pools), were selected as target genes.

### 2.7. Functional Validation and Phylogenetic Analysis of Target Genes

Primers were designed based on the target gene sequences, and PCR amplification was performed using DNA from individual strains of the segregated population. The amplified products were sequenced to further validate the association between SNP mutation sites and monokaryotic fruiting traits.

Protein sequences from representative fungi species were retrieved from NCBI (https://www.ncbi.nlm.nih.gov/, accessed on 18 January 2025) using target gene sequences as a query for phylogenetic analysis. Multiple sequence alignment was performed using the ClustalX software (http://www.clustal.org, accessed on 18 January 2025)) with default parameters. A phylogenetic tree was constructed using MEGA 7, using the maximum likelihood (ML) method with the JTT substitution model, supported by 500 bootstrap replications.

## 3. Results

### 3.1. Construction of a Genetic Segregation Population for Monokaryotic Fruiting Control Genes

A total of 80 FF002 monokaryotic strains were obtained through basidiospore collection and monokaryotic strain preparation. Among these, 60 strains (75%) exhibited monokaryotic fruiting trait, while the remaining 20 strains (25%) were non-fruiting, with no other phenotypic variations observed (Appendix A).

Monokaryotic fruiting and non-fruiting strains of FF002 were selected for selfing. All 47 monokaryotic strains derived from the S_f1_ (43 × 56) successfully produced fruiting bodies, and monokaryotic strains from three S_f2_ (1 × 4, 3 × 12, and 19 × 40) generations also consistently formed fruiting bodies. These results indicate that the S_f1_ (43 × 56) and S_f2_ (1 × 4, 3 × 12, and 19 × 40) generation strains were homozygous for the monokaryotic fruiting trait. Among the 53 monokaryotic strains derived from the S_n1_ (36 × 62), 15 formed fruiting bodies, whereas 38 did not. Three S_n2_ (5 × 14, 10 × 11, and 13 × 25) generations were established from the non-fruiting monokaryotic strains. Notably, all 33 monokaryotic strains from the S_n2_ (5 × 14) failed to produce fruiting bodies. Furthermore, monokaryotic strains from three S_n3_ (2 × 9, 3 × 4, and 8 × 32) generations derived from 5 × 14 also consistently failed to form fruiting bodies. These findings indicate that the S_n2_ (5 × 14) and S_n3_ (2 × 9, 3 × 4, and 8 × 32) generation strains were homozygous for the monokaryotic non-fruiting trait (Figure 2, Table A1, Appendix A).

A dikaryotic strain S_f*n_ 30 × 8 was constructed by crossing the fruiting monokaryotic strain (FF002-PL-30F) from S_f2_ 19 × 40 with the non-fruiting monokaryotic strain (FF002-PL-8N) from S_n3_ 8 × 32. In addition, 156 monokaryotic strains were collected and established as a genetic segregation population for studying monokaryotic fruiting control genes. Of these, 74 strains (FF002-F) produced monokaryotic fruiting bodies, while 82 did not (FF002-N) (Figure 2, Appendix A). The observed 1:1 segregation ratio (χ^2^ test, χ^2^ = 0.41, *p* > 0.05) suggests monogenic control of this trait.

To identify candidate genes associated with fruiting, we constructed two bulked segregant populations (BSA pools)—each comprising 24 monokaryotic strains (FF002-F and FF002-N)—for high-throughput sequencing library and performed BSA-seq analysis. Strains of two parental lines (FF002-PL-30F and FF002-PL-8N) were collected for further library construction and BSA-seq.

### 3.2. Identification of Monokaryotic Fruiting Control Genes Through BSA-Seq Analysis

After trimming the adapter and removing low-quality data, 20.4–30.7 Gb of high-quality clean data were maintained per library. Alignment to the reference genome yielded uniquely mapped reads ranging from 30.5% to 38.8%, with an average sequencing depth of 68.80 X (Table A3).

A total of 7021 SNPs and 292 InDels with high-quality between two parent lines (fruiting and non-fruiting lines) were selected for association analyses. These variants were evaluated using four complementary methods: Δ(SNP-index), Euclidean distance (ED), G-value, and Fisher’s exact test. With a 99% confidence threshold, all four analytical methods consistently pinpointed a significant association between scaffold19 (45,000–55,000) and the monokaryotic fruiting trait, establishing this region as the candidate (Figure 3E).

Five candidate genes were identified within the target genomic interval, based on gene annotation analysis (Figure 3F, Table 1). Among them, a total of 12 non-synonymous mutations were detected within the coding sequence (CDS) regions by integrating SNP data from resequencing data of segregated pools (Table A2). Intriguingly, the *FV-L110034160* gene contained a mutation 121 bp downstream of the ATG start codon in its third exon, with the two parental lines displaying homozygous reference (0|0) and homozygous variant (1|1) genotypes. This mutation was consistently reflected in the dominant alleles of the two gene pools, corresponding to the parental genotypes. This T→G transversion contributed to a substitution from a hydrophilic serine (Ser) to a hydrophobic alanine (Ala) (Figure 3G). The *FV-L110034160* gene spans 2193 bp, containing seven exons and six introns, with a 1710 bp coding sequence (CDS) that encodes a 569-amino acid protein. This protein functions as a component of the U2 small nuclear ribonucleoprotein (snRNP) complex, which mediates precursor mRNA (pre-mRNA) splicing. The protein contains a conserved SF3B2 domain (also designated SAP145 or SF3b145), a key element of the SF3b complex that facilitates intron excision during U2 snRNP-dependent pre-mRNA splicing. This finding aligns with previous studies reporting significant upregulation of RNA splicing-related genes during fruiting body formation in *F. velutipes* [38] and *Cyclocybe aegerita* [12]. Collectively, these results indicate a significant association between *FV-L110034160* and monokaryotic fruiting, supporting it as a strong candidate gene governing this developmental trait.

### 3.3. Validation of the Target Gene and Phylogenetics Analysis

Using primers specific to *FV-L110034160*, we performed PCR amplification and sequencing of individual strains from both gene pools. The results revealed complete genotype–phenotype correlation: all non-fruiting monokaryotic strains were consistent with the FF002-PL-8N parental genotype (T) at the SNP locus scaffold19-47,775-T-G, while fruiting monokaryotic strains matched the FF002-PL-30F parental genotype (G) (Figure 4). This alteration likely contributes to the loss of function in the conserved domain of the encoded protein and is associated with the phenotypic change in the fruiting trait of monokaryotic strains, suggesting that *FV-L110034160* may be a candidate gene associated with the monokaryotic fruiting trait.

Phylogenetic analysis of the *FV-L110034160* gene revealed distinct evolutionary patterns. The protein showed high conservation among species within the same genus, such as *Armillaria borealis* and *Armillaria mellea* within the genus *Armillaria*, as well as *Amanita muscaria* and *Amanita rubescens* within the genus *Amanita*. However, significant divergence was observed at taxonomic levels above the genus, particularly between *F. filiformis* and *Lentinula edodes* (Figure 5). The conservation of such genes among edible fungi species suggested their important roles in the development of the fruiting body.

## 4. Discussion

The mechanism of fruiting body formation has long been a focal point in edible fungi research, with investigators employing multiple perspectives, including physical environment, nutritional physiology and mutant analyses, to identify key regulatory genes. Among these approaches, the use of mutant strains defective in fruiting body formation, among model species, has proven to be one of the most critical and effective strategies. Mutant-based approaches have successfully identified several key genes involved in fruiting body formation, including the blue light receptor genes *wc-1* and *wc-2* [39]; the development switch gene *Cc.rmt1* [40], which encodes arginine methyltransferase, and, when regulated, leads to sclerotia formation instead of fruiting bodies following hyphal knotting; the inhibit fruiting body formation and dikaryotic hyphal growth gene *Cc.snf5* [41]; the regulatory stipe elongation gene *eln2* [4]; the influence clamp connection and fruiting body formation genes *clp1* [42], *Poclp1* [43], *Fvclp1* [44], and *ich1* [45]; and the prevent primordia knotting gene *thn* [46]. Thus, studying mutants defective in fruiting body formation in model species has proven to be an effective strategy for identifying key control genes. However, this approach is constrained by its reliance on the detection of dominant mutations and the randomness of mutation sites, which significantly hinders the elucidation of fruiting body formation mechanisms. Therefore, increasing the diversity of key control genes for fruiting body formation and transitioning from studies limited to dominant mutations to those capable of identifying both dominant and recessive mutations represent critical directions for advancing our understanding of fruiting body formation in edible fungi.

Monokaryotic fruiting refers to the phenomenon in which monokaryotic strains of basidiomycetes with a heterothallic sexual reproductive system form fruiting bodies without mating. This trait has been observed in over 30 basidiomycete species [47]. In addition to *F. filiformis* [25,26], common edible fungi species such as *Auricularia heimuer* [48], *Agrocybe aegerita* [49], and *Pholiota microspora* [50] also exhibit monokaryotic fruiting. Since monokaryotic strains possess only one set of chromosomes, both dominant and recessive mutations in key fruiting body control genes can disrupt fruiting body formation, thereby expanding the range of genes amenable to study. Consequently, the phenomenon of monokaryotic fruiting provides a novel approach for investigating key control genes by directly correlating phenotypic traits with genotypes through eliminating interference from allelic variations. This trait offers significant advantages for studying key control genes involved in fruiting body formation.

Building on the identification of control genes for fruiting body formation in *F. filiformis* through monokaryotic fruiting trait analysis, we further employed multiple rounds of selfing purification to generate homozygous strains with highly uniform genetic backgrounds, differing only in the genetic loci controlling monokaryotic fruiting. This approach minimized interference from genetic variability. Our experiments revealed that homozygous strains for the fruiting trait could typically be obtained after 2–3 generations of selfing, and in some cases, after just one generation, are far fewer than the generations required in plants. This efficiency likely stems from the direct observability of the monokaryotic fruiting trait, which allows targeted selection of monokaryotic strains with identical fruiting phenotypes for selfing crosses, significantly reducing the time required for genetic manipulation. Moreover, since the segregation population was derived from continuous selfing of a single starting strain, the genetic background remained highly consistent. This enabled direct localization of candidate genes to a 10 kb interval using BSA-seq alone, eliminating the need for subsequent development of molecular markers for fine-mapping and achieving a mapping resolution comparable to map-based cloning methods [51,52]. However, challenges arose during selfing, including the accumulation of deleterious mutations and segregation distortion. By the S3 generation, the probability of successful mating among monokaryotic strains significantly decreased, posing substantial difficulties for applying this technique to map quantitative traits controlled by multiple genes (as selfing may become infeasible before major-effect genes are purified). These limitations warrant further investigation.

The mapping rate of reads to the reference genome ranged from 30~38%, which is relatively low for fungal resequencing studies. To investigate potential causes, we aligned our resequencing data to a high-quality reference genome of *F. filiformis* strain KACC42780 (GenBank: GCA_000633125.1), noted for its superior assembly. However, the mapping rates remained modest, ranging from 35.83~46.55%. This result suggests that reference genome incompleteness is unlikely to be the primary factor contributing to the low mapping rates. Instead, we hypothesize that the low mapping rates may result from substantial genetic divergence between the wild, undomesticated *F. filiformis* strain used in this study and cultivated strains. This interpretation is consistent with findings by Zhao et al. [53], who reported significant genomic differences between wild and domesticated *F. filiformis* populations, including variations in genome size and gene content. While the lower mapping rates could potentially lead to the oversight of additional loci closely associated with monokaryotic fruiting traits, we conducted rigorous statistical analyses, including Δ(SNP-index), Euclidean distance (ED), G-value, and Fisher’s exact test, to validate the identified loci. These analyses consistently demonstrated a significant correlation between the mapped loci and monokaryotic fruiting traits, suggesting the accuracy of our results.

Although this study successfully identified the candidate gene *FV-L110034160* associated with monokaryotic fruiting in *F. filiformis* through selfing purification and BSA-seq analysis, and established a strong correlation between SNPs in this gene and the observed phenotype. The protein encoded by *FV-L110034160* functions as a component of the U2 small nuclear ribonucleoprotein (snRNP) complex, playing a critical role in pre-mRNA splicing as a core constituent of the spliceosomal U2 snRNP. This component is highly conserved across eukaryotes [54]. During fruiting body formation, precise temporal regulation of gene expression is essential, and the spliceosome is pivotal for post-transcriptional regulation, including alternative splicing [55]. Studies have demonstrated that mutations in U2 snRNP components in *Candida albicans* lead to defects in hyphal morphogenesis [56]. Similarly, in *Saccharomyces cerevisiae*, the absence of U2 snRNP-related proteins directly impairs late-stage development [57]. Moreover, significant differential expression of U2 snRNP components has been observed during fruiting body formation in *F. filiformis* [38] and *C. aegerita* [12]. However, the direct impact of this gene on monokaryotic fruiting body formation has not yet been validated. In future studies, the gene candidates of this research will be further validated through CRISPR and RNAi techniques. Additionally, building on our efficient target gene mapping pipeline—which combines trait-specific selfing purification, genetic population construction, BSA-seq, and candidate gene validation—we will investigate genes controlling diverse monokaryotic fruiting traits (e.g., primordia presence/absence, differentiation extent, and morphological characteristics) across additional *F. filiformis* strains. Furthermore, we plan to employ transcriptome sequencing to analyze the mycelial and primordium stages of the *F. filiformis* strain FF002. This approach aims to identify candidate genes associated with fruiting body formation, thereby providing deeper insights into the mechanisms underlying fruiting body formation in *F. filiformis*. This systematic approach will help construct a comprehensive regulatory network of fruiting body development in *F. filiformis*, advancing both mechanistic understanding and practical applications in edible fungi research.

## 5. Conclusions

This study successfully established *F. filiformis* as the model organism to identify genes related to fruit body formation. By employing standardized growth conditions, genetic population construction, and BSA-seq analysis, we identified *FV-L110034160*, a key gene encoding a U2 snRNP component critical for pre-mRNA splicing. A functional SNP (scaffold19-47,775 bp, T→G) causing a Ser→Ala substitution was found to alter gene function, directly influencing monokaryotic fruiting. PCR-based validation confirmed the mutation’s phenotypic association. These findings not only advance our understanding of fruiting body development in *F. filiformis* but also provide a methodological framework for studying similar mechanisms in other edible fungi. This work further offers potential strategies for improving fruiting body induction in economically important basidiomycetes that remain challenging to cultivate artificially. Future studies should explore the broader regulatory network involving *FV-L110034160* and assess its applicability across fungal species.

## Figures and Tables

**Figure 1 jof-11-00512-f001:**
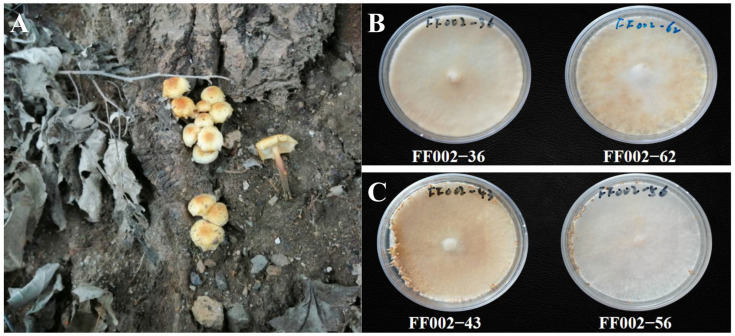
Fruiting bodies and monokaryotic strains of *F. filiformis* strain FF002. (**A**) Wild *F. filiformis* strain fruiting bodies; (**B**) non-fruiting monokaryotic strains; (**C**) fruiting monokaryotic strains.

**Figure 2 jof-11-00512-f002:**
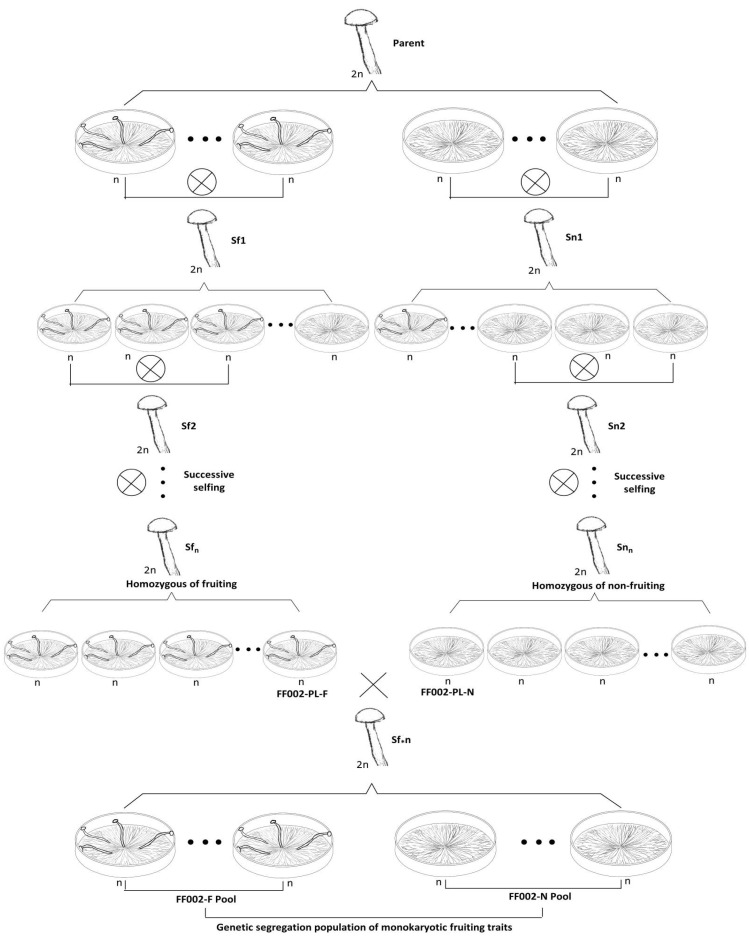
Process of constructing a genetic segregation population for monokaryotic fruiting traits.

**Figure 3 jof-11-00512-f003:**
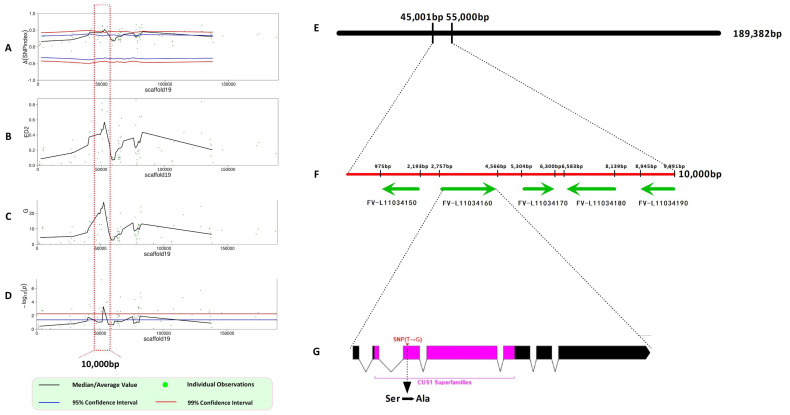
Identification of the monokaryotic fruiting control genes. (**A**) Manhattan plot showing the distribution of Δ(SNP-index) on scaffold19; (**B**) Manhattan plot showing the distribution of Euclidean distance (ED2) on scaffold19; (**C**) Manhattan plot showing the distribution of G-value on scaffold19; (**D**) Manhattan plot showing the distribution of log-transformed Fisher’s exact test *p*-value distribution –log10(p) on scaffold19; (**E**) physical map of scaffold19; (**F**) the 9.8 kb region of the monokaryotic fruiting region based on four independent BSA-seq mappings containing 5 predicted genes according to the FV-L11 reference genome; (**G**) schematic illustration and characterization of *FV-L110034160*, exons and introns are shown in boxes and lines, respectively. The mutation site was indicated.

**Figure 4 jof-11-00512-f004:**
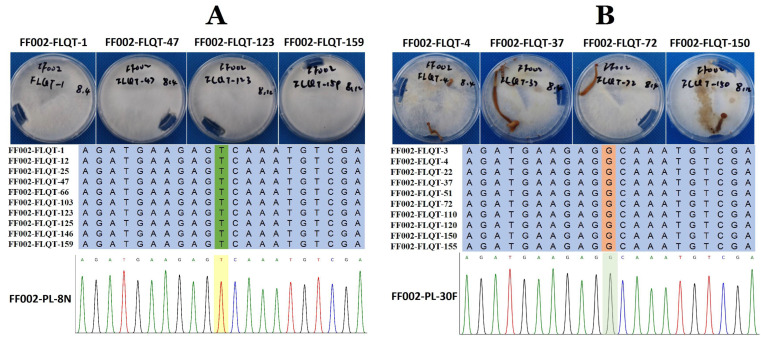
Validation of the *FV-L110034160* gene of fruiting and non-fruiting monokaryotic strains. (**A**) Morphological characteristics and DNA sequences near the mutation site of non-fruiting monokaryotic strains in the genetic population, alongside sequencing chromatograms of the mutation site in non-fruiting monokaryotic parental strain FF002-PL-8N used for population construction (in the genetic population, the phenotype of monokaryotic non-fruiting strains consistently exhibits a T base at the mutation site, identical to the monokaryotic non-fruiting parental strain); (**B**) morphological characteristics and DNA sequences near the mutation site of fruiting monokaryotic strains in the genetic population, alongside sequencing chromatograms of the mutation site in fruiting monokaryotic parental strain FF002-PL-30F used for population construction (in the genetic population, the phenotype of monokaryotic fruiting strains consistently exhibits a G base at the mutation site, identical to the monokaryotic fruiting parental strain).

**Figure 5 jof-11-00512-f005:**
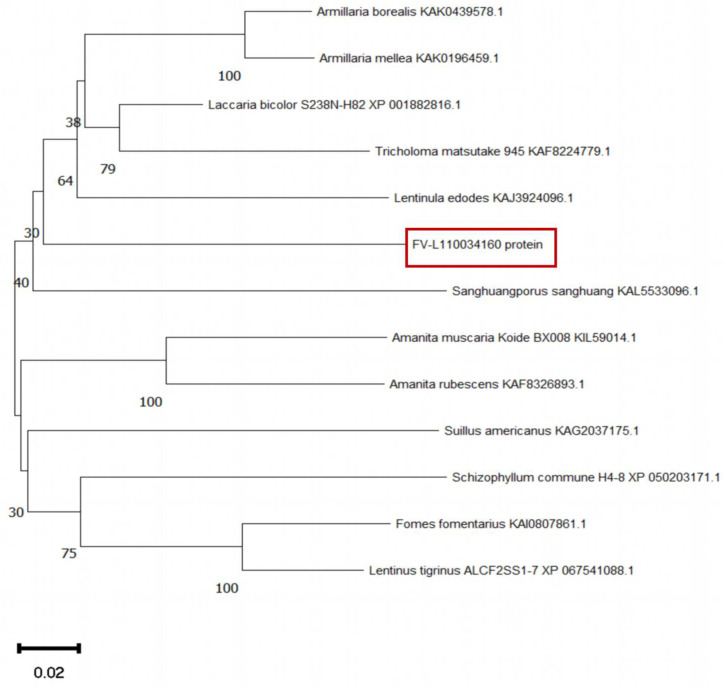
Phylogenetic tree of mushroom DUF382-domain-containing protein. The tree for DUF382-domain-containing protein homologue amino acid sequences was calculated and constructed using the MEGA program (The protein encoded by the target gene was highlighted with a red square frame).

**Table 1 jof-11-00512-t001:** Five candidate genes screened out by annotation information.

Gene ID	Location	PFAMs	Pathway	Function Description
*FV-L110034150*	Scaffold19 (45,969–47,338)	ASXH	-	Asx homology domain
*FV-L110034160*	Scaffold19 (47,374–49,566)	DUF382 (CUS1)	ko03040 (pentose phosphate pathway)	DUF382-domain-containing protein
*FV-L110034170*	Scaffold19 (49,778–51,300)	BCDHK_Adom3, HATPase_c	-	Dehydrogenase
*FV-L110034180*	Scaffold19 (51,399–52,780)	3Beta_HSD, Epimerase, NAD_binding_4	-	Dihydrokaempferol 4-reductase activity
*FV-L110034190*	Scaffold19 (53,799–55,299)	HAD_2	ko00561 (glycerolipid metabolism), ko01100 (metabolic pathways)	HAD-hyrolase-like

## Data Availability

The original contributions presented in this study are included within the article/Appendix A. Further inquiries can be directed to the corresponding authors.

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
