# Peer review of "Identification of Critical Candidate Genes Controlling Monokaryon Fruiting in Flammulina filiformis Using Genetic Population Construction and Bulked Segregant Analysis Sequencing"

_jof, 2025, doi:10.3390/jof11070512_

Round 1

Reviewer 1 Report

The BSA-Seq analysis performed is an interesting approach to elucidate key genes involved in monokaryotic fruiting in Flammulina filiformis. The research is well designed, and conclusions are supported by the results.

As a commentary, although the contribution to understanding the fruiting body formation in this species seems that not only a punctual mutation in a single gene could have absolute effect in the loss of fruiting capability. It should be interesting to complement this approach with a transcriptomic analysis to identify candidate genes directly involved in fruiting boy formation as well as regulatory genes.

Please improve the quality of samples selected for the pictures in the figure 4A and B. cultures in petri dishes seem to be used previously since a small piece of colonies is lacking in all of them.

Also, the handwritten labels on petri dishes (figure 1B, C and figure 4 A,B) are hard to read. Please replace them.

Reviewer 2 Report

The manuscript presents a well-designed and rigorously executed study that establishes Flammulina filiformis as a model organism for investigating genes associated with fruiting body formation. The authors successfully employed standardized cultivation conditions, constructed a genetic population, and conducted BSA-seq analysis to identify key genetic loci. The experimental approach is robust, the results are well-validated, and the findings have clear implications not only for the winter mushroom industry but also for broader applications in breeding programs for other mushroom species. 

I have only a few minor comments:

1. The title should avoid abbreviations such as “BSA-seq” to ensure accessibility and clarity for a wider readership.

2. What is QTL interval (line 156)?

3. In Figure 3 (A–D), black, red, and blue lines, as well as green dots, are shown without explanation. Please add a legend to indicate their meanings.

4. The abstract states that FV-L110034160 is located on scaffold11, but in the Results section, it is reported on scaffold19. This discrepancy should be corrected.

5. The Discussion section highlights several known genes related to fruiting body development (e.g., wc-1, clp1, ich1), but fails to discuss the potential role of FV-L110034160, which encodes a U2 snRNP complex component. Given this gene is central to your findings, a more informative discussion is needed regarding its role in pre-mRNA splicing and how this might relate to fruiting body formation.

6. Pooling DNA samples before BSA-seq can introduce biases and affect variant calling accuracy. A brief discussion of this limitation should be included.

Reviewer 3 Report

This study presents a well-executed and clearly structured research on gene indentification related to monokaryon fruiting in Flammulina filiformis, using genetic population constructon and BSA-Seq approach. The authors succsessfully narrow down a candidate region to a 10kb interval on scaffold19 and identify a SNP within FV-L110034160, which encodes a component of the U2 snRNP complex.
The manuscript offers novelty and significance, esp. in terms of utilizing monokaryotic systems for genetic dissection, which are less commonly exploited compared to dikaryotic models. The combination of selfing purification and BSA-sequencing appears both efficient and potentially scalable to other traits.
However, some parts require further clarificcation and refinement. In particular, overstated claims regarding gene function should be moderated. Additionally, inclusion of comments on missing expression data and mapping efficiency will help strengthen the manuscript.

Major Points
1. While a strong SNP-to-phenotype correlation is established, there is no direct evidence (e.g., knock-out, knock-in, or RNAi experiments) that the SNP in FV-L110034160 causally affects monokaryotic fruiting. Please revise statements such as "establishing it as a principal candidate gene" to "supporting it as a strong candidate gene". Also, acknowledge this limitation explicitly in the Discussion.

2. The mapping rate of reads to the reference genome ranged from ~30–38%, which is relatively low for fungal resequencing studies. Provide a short discussion of potential reasons (e.g., reference genome incompleteness, genetic divergence of the strain) and its possible impact on variant calling accuracy.

3. There is no transcriptomic or qPCR data to confirm whether FV-L110034160 is differentially expressed between fruiting and non-fruiting strains. While not essential for this manuscript’s publication, the authors should mention in the Discussion that such analyses would strengthen the functional interpretation of the identified variant.

Minor Points
1. Phrases such as "critical regulatory gene" or "drives the phenotypic change" are too definitive without functional assays. Use more tentative language, e.g., “likely contributes to” or “is associated with”.

2. The mutation site in Figure 4 is important, but the figure legend and in-text explanation are minimal. Expand the figure legend to clarify the genotype-phenotype correlation and what the chromatograms show.

3. Several sections would benefit from minor grammatical and stylistic improvements. For example: 
 - “making significant challenging to identify” → “making it significantly challenging to identify”
 - “re-sequencing” → “resequencing” (standard spelling)

4. Duplicate citations: Takagi et al. (2013) appears twice as Ref 17 and Ref 35.
